# Research on the Mechanism of Multi-Domain Coupling Centrifugal Electrostatic Blowing Flying Deposition

**DOI:** 10.3390/mi13091378

**Published:** 2022-08-24

**Authors:** Guojie Xu, Bei Zhou, Jian Guo, Jun Zeng, Rongguang Zhang, Nian Cai, Yongxing Li, Peixuan Wu, Xun Chen, Han Wang, Juan M. Ruso, Zhen Liu

**Affiliations:** 1State Key Laboratory of Precision Electronic Manufacturing Technology and Equipment, Guangdong University of Technology, Guangzhou 510006, China; 2Guangdong Provincial Key Laboratory of Micro-Nano Manufacturing Technology and Equipment, Guangdong University of Technology, Guangzhou 510006, China; 3Foshan Nanofiberlabs Co., Ltd., Foshan 528225, China; 4Soft Matter and Molecular Biophysics Group, Department of Applied Physics, University of Santiago de Compostela, 15782 Santiago de Compostela, Spain; 5Department of Physics and Engineering, Frostburg State University, Frostburg, MD 21532, USA

**Keywords:** nanofibers, centrifugal electrostatic injection, flight deposition prediction, MATLAB optimal solution, COMSOL simulation

## Abstract

The centrifugal electrostatic blowing process proposed in this paper solves the difficult continuous and stable deposition problem in the traditional centrifugal electrostatic spinning process. By establishing a flight deposition model of the centrifugal electrostatic spraying process, CFD is used to simulate and analyze the electrohydrodynamic effect of centrifugal jets, and the driving mechanism is explored. Subsequently, MATLAB is used to obtain the optimal solution conditions, and finally, the establishment of a two-dimensional flight trajectory model is completed and experimentally verified. In addition, the deposition model of the jet is established to clarify the flight trajectory under the multi-field coupling, the stable draft area of the jet is found according to this, and the optimal drafting station is clarified. This research provides new ideas and references for the exploration of the deposition mechanism of the centrifugal electrostatic blowing and electrostatic spinning process.

## 1. Introduction

Electrospinning technology is one of the main methods for preparing nanomaterials, especially inorganic multilevel nanomaterials [1]. By controlling the parameters of the electrospinning process, nanofibers with a variety of morphologies, including nanowires, nanotubes, nanobelts, and nanorods, can be synthesized [2,3,4,5,6]. The receiving electrode plates is one of the important components of the electrospinning system. The improved receiving electrode plate of barrel shape and disc shape are used to synthesize the electrospinning fiber with the same orientation, which expands the applications of electrospinning [7,8,9]. In order to increase the output of the electrospinning, the multi-nozzle electrospinning method is most commonly used in the current industry [10,11]. However, with the development of the nanofiber industry and the upgrade of materials, the complexity of the combination of raw material solutions has gradually increased in order to meet industry performance requirements. The polymer materials with complex blending rheological properties have different phase transition processes during the spinning process, which are prone to defects such as spinning electrode crystals, gels, and hanging filaments [12,13,14,15,16,17,18]. Therefore, it becomes difficult to meet the increasingly complex material preparation requirements with the nanofiber preparation process that uses Coulomb force as a single source of field force [19].

With further research, in order to overcome the shortcomings of traditional electrospinning and meet different types of needs, nanofiber preparation systems such as solution jets, bubble spinning, melt micro-spinning, and centrifugal spinning have been established [20,21,22,23]. It is found that centrifugal electrospinning has great advantages in the preparation of fluffy and porous three-dimensional nanofiber network structures [24]. By introducing an electrostatic field, the fiber can be further refined, and the drafting effect is improved. Zhang and others summarized the influence of equipment parameters and polymer solution properties on the morphology of nanofibers in the centrifugal spinning process [23]. However, the establishment of predictive models and the changing laws of nanofibers, jet dynamic tension, and spatial motion are still relatively vacant. Few researchers have established the relationship model between process parameters, geometric parameters, and nanofiber diameters. There are few optimization methods for various parameters, and it is difficult to provide a theoretical basis for efficient and clean nanofiber preparation. Noroozi used experimental and mathematical modeling results to characterize, and obtained the influence of factors such as rotation speed and environmental humidity on the performance of centrifugal electrospinning [25]. However, the influence of many parameters, such as surface tension, evaporation rate, and rheological effects, is still unclear, and further research and exploration are still needed. In 2021, Bülin Atici studied and summarized the structure, characteristics, and performance of centrifugal spinning fiber, and observed that if more research is conducted on the modeling and prediction of fiber diameter and morphology, it will be helpful in further improving the applications of centrifugal spinning fiber in different fields [26]. In summary, in order to improve the preparation of centrifugal electrospinning, although many scholars have explored the setting of its flow channel, field force application, control and other parameters, it is necessary to achieve continuous and stable preparation. The coupling of the centrifugal effect and the electric field is still very challenging [27,28,29,30]. At present, the industry has not yet been able to efficiently propose a stable drafting method for the centrifugal charged jet and the preparation process of a wide range of adaptable fibers. The solution is to establish a multi-domain coupling centrifugal electrostatic blowing flying deposition.

Aiming at the above-mentioned problems, this paper proposes the following centrifugal electrostatic spraying scheme: First, the flight trajectory in the centrifugal electrospinning process is modeled and analyzed, the layout of the drafting station is clarified, and the constraint conditions of the airflow aerodynamic parameters are determined. Then, based on the motion constraints, Euler equations, and continuity equations, a stable jet flight deposition model is established to predict the flight trajectory under the coupling of multi-domain parameters. Based on this, we find the jet-stream stable drafting zone. Finally, the flight deposition trajectory model is verified by experiments, and the actual trajectory result is determined to be consistent with the result obtained from the mathematical model, which can be used for basic analysis and prediction of the jet trajectory.

## 2. Analysis of the Forming Mechanism of the Centrifugal Electrostatic Blowing Process

### 2.1. Design and Numerical Simulation of Centrifugal Electrostatic Blowing Experiment Platform

The centrifugal electrostatic blowing system is mainly composed of four parts: rotation, liquid supply, high-voltage power supply, and air blowing, as shown in Figure 1. The main design idea of the scheme is to load the air jet nozzle on the basis of the traditional centrifugal electrospinning process, and to use the jet nozzle to longitudinally draw the equipment ejected by the centrifugal rotor to form a collection at the receiving end. In addition, the selection and design of the connection mode of the high-speed rotating part and the high-voltage fixed end ensure the effectiveness of the application of the electric field drafting force.

This paper uses the k−ε transient state of multiphysics simulation in COMSOL, version 5.6, accessed in 2020. simulation to solve the centrifugal electrospinning jet model, and establishes a mathematical model for the fully developed turbulent flow state [31]. The simulation boundary conditions of the motion state of the polymer jet under the action of the centrifugal electrostatic multiphysics field are shown in Figure 2.

Numerical simulation results show that the rotation speed plays a leading role in the jet motion state. The simulation results at different rotation speeds are shown in Figure 3. Due to the limited convergence interval of COMSOL, continuous addition of boundary conditions or expansion can easily cause the model to diverge. In order to further clarify the long-distance flight deposition trajectory of the jet in a two-dimensional plane, it is necessary to establish a complete mathematical model of the jet during the centrifugal spinning process with the rotational speed as the main object to determine the predictability of the flight deposition trajectory.

### 2.2. Model Construction of Centrifugal Electrostatic Blowing Process Based on Centrifugal Rotor Platform

Based on the two-dimensional model of the centrifugal rotor, the Cartesian coordinate system is established with the center of the rotor as the coordinate origin, as shown in Figure 4.

In order to reduce the amount of calculation, this section degenerates the actual model. The rotor is set to have only one spinning port, and it is represented by point O in the coordinate system, that is, this point is the liquid outlet position of the rotor connector. Considering only the centrifugal force, and treating the jet motion as a plane motion, any point on the jet can be described as $\;\left(X\left(s,t\right)+S_{0},\;0,\;Z\left(s,t\right)\right)$. In order to facilitate the characterization of all the micro-elements on each jet interface, this paper selects the jet center trace to establish a polar coordinate system. The polar coordinates of the jet cross-section are shown in Figure 5.

With the aid of this coordinate system, through the three-unit normal vectors, the derivation of the jet field is described as:(1)$$u=ue_{s}+ve_{n}+we_{\emptyset }$$

In order to derive the differential equation of the fluid, the order of magnitude comparison method is used to estimate the middle order of the complete conservation equation based on the flow characteristics, and the small order term is omitted to obtain a direct analytical solution.

It should be noted that incompressible N-S equations are used here. The Navier-Stokes equations are at the heart of fluid flow modeling. Solving these equations under specific boundary conditions, such as inlets, outlets, and walls, predicts fluid velocity and pressure in a given geometry and tracks interfaces between immiscible fluids [32]. The Navier-Stokes equations are equations used to describe the motion of fluids and can be seen as Newton’s second law of fluid motion [33]. The Navier-Stokes equations express the conservation of momentum, while the continuity equations express the conservation of mass.
(2)∇·u=0
(3)$$\rho \left(u\cdot \nabla \right)u=\nabla \;\left[\mu \;\left(\;\nabla u+{\left(\nabla u\right)}^{T}\;\right)\right]+\rho g-\nabla P$$

In the formula *u*, *μ*, *P*, *ρ*, and *g* represent velocity, viscosity, pressure, density, and gravity, respectively.

The centrifugal jet motion constraint equation is as follows:(4)$$\frac{D}{Dt}\;\left(R\left(s,\emptyset ,t\right)-n\right)=0$$

However, due to the small size of the jet in the centrifugal jet movement, conventional simplified processing methods are not applicable. For the jet flow that can be regarded as a steady state, having no internal heat source, being two-dimensional, and constant in the centrifugal jet under the influence of the body force, the solution of the jet is suitable for solving the boundary layer differential equation of the viscous fluid with complex variables. The matching gradual progress method (the method of matched Asymplotic Expansions). The following will directly perform non-dimensionalization in order to directly substitute it into the aforementioned basic fluid equation. The schematic diagram of the velocity of the jet in the centrifugal rotation process is shown in Figure 6, and the jet trajectory solved by the fourth-order Runge-Kutta method is shown in Figure 7.

### 2.3. Simulation Experiment Analysis Based on Numerical Model of Centrifugal Electrostatic Blowing Process

In this paper, the conventional fourth-order Runge-Kutta methods, namely the ODE45 function in MATLAB, is used to solve the mathematical model as shown in Figure 8. The jet trajectory set obtained by the solution has a large deviation from the actual expectation and cognition, and then the ODE15i function is used to solve the centrifugal jet model.

The jet trajectory obtained by using different Weber numbers under the implicit Runge-Kutta method is shown in Figure 9.

From Figure 8, it can be observed that using the implicit Runge-Kutta methods ODE15I, under the premise that the iterative Jacobian matrix cannot be determined, different iteration precision settings have a great impact on the iteration effect. In Figure 8, the relative accuracy of the iteration is set from 1 × 10^−3^ to 1 × 10^−6^. When the iteration accuracy is too low (1 × 10^−1^ to 1 × 10^−2^) and too high (1 × 10^−7^ to 1 × 10^−8^), the graph deviates too much from the predicted trajectory. Therefore, with this method, if the iteration accuracy is set too low, it will deviate too far from the standard at the beginning of the iteration, and if it continues to iterate, there will be a large deviation, and it is difficult to meet the iteration accuracy.

According to the iterative results in Figure 8, comprehensively evaluate the computational complexity and solution accuracy, and choose ‘RelTol’ = 1 × 10^−5^, ‘AbsTol’ = 1 × 10^−6^ as the iterative conditions, and continue to complete the solution for the motion states of different solution parameters. The results are shown in Figure 10. In the figure, the fluid Weber number We is set to 10, and as the liquid Rossby value Rb increases from 2 to 10, the exit angle of the jet out of the rotor is larger, and the radius of rotation is also larger. It is more susceptible to centrifugal outward movement under the action of inertial force, so it has a larger movement radius. This is mainly because the Rossby value Rb is the ratio of the characteristic speed to the characteristic length, which can be used to characterize the effect of the inertial force and the Coriolis force. The greater the value of Rb, the greater the effect of inertial force, the more the jet has a tendency to move outwards, and the larger the radius of movement. Therefore, the numerical model is consistent with the theoretical prediction.

When the We number is relative to the Rb number, the jet trajectory does not change significantly. As in Figure 10, in the interval of 0 < We < 30, as the We number increases, the jet has a greater tendency to move outwards and a larger radius of motion. However, if the We value continues to increase, the change trend of the jet trajectory decreases. The Weber number mainly represents the dimensionless value of the inertial force and the surface tension effect. The smaller the Weber number, the greater the influence of the surface tension on the jet, and vice versa, the greater the influence of the inertial force on the jet. When We = inf, the jet has no surface tension. The solution reveals that when We reaches a certain level, the outward trend of the jet will decrease.

## 3. Experimental Verification and Analysis

### 3.1. Experimental Platform Design and Implementation

In order to monitor the movement of the high-speed jet in real time, this paper uses a camera and a stroboscope to observe the trajectory of the centrifugal jet. The experimental platform is shown in Figure 11. The stroboscopic light source is mainly provided by the stroboscope, and the image is collected by a microscopic camera placed directly above the rotating cup.

For centrifugal spinning, it is difficult to achieve shooting with a camera alone due to a combination of the fast jet motion speed and the thin jet stream. To capture a stable jet and achieve continuous observation, a high-speed microscope camera with a large depth of field must be used. However, the depth of field limit of this type of camera is relatively large, and it is expensive. Moreover, it is difficult to apply in the field of chemical fiber industrial production. Therefore, in this study we use a stroboscopic light source combined with ordinary microscope camera to detect. With this method, it is not limited by the small depth of field of the microscopic camera, and greatly reduces the requirements for the camera. The components of the observation platform are shown in the Table 1.

The jet flow observation based on this method is due to the strong repetitiveness of the jet state. For yaw motion, rotational motion, or oscillating motion, a fast-flashing stroboscopic light source can produce image overlap when the stroboscopic frequency overlaps with the jet motion frequency. Such overlap can be observed by the human eye or the camera at a lower frequency of photography. Therefore, from another point of view, if the motion picture of the jet can be detected by the stroboscope, this proves that there is a clear trajectory in a certain period of this type of motion.

### 3.2. Experimental Results and Analysis

This article is suitable for a spinning experiment with PEO (Polyethylene oxide) solution in a specific concentration spinning zone. Due to its relatively large molecular weight (4,000,000 g/mol), the range of mass fraction of the configuration is set from 4.5% to 7.5%. Its viscosity is also larger, increasing from 2786 to 20,200 m, and its physical property test results are shown in Table 2.

The change of the jet with a mass fraction ranging from 4.5% to 7.5% is shown in Figure 12.

It is observed from the jet trajectory that, under the Coriolis force along the radial direction to the center of the circle, the jet bends to form a spiral trajectory after it exits the needle. In order to quantify the influence of jet parameters on the trajectory, it is necessary to analyze the movement angle of the jet after it flows out of the needle. The distance is measured from the origin of the jet when the rotation angle reaches φ = 2π and compared with the model fitting curve obtained in the previous section.

The trajectories of solutions with different Weber numbers after being shot and rotated through the rotor cup are shown in Figure 13. When the Weber number is small, that is, when 1 < We < 25, as the Weber number increases, D increases more obviously. When the Weber number is greater than 30, as the Weber number increases, the increasing trend of D slows down, and there is a tendency to converge around 12 mm. These two trends appear simultaneously in the actual experimental trajectory results and the results obtained from the mathematical model in the previous section, and can be used for the basic prediction of jet trajectories.

## 4. Conclusions

This article explores the drafting mechanism of airflow on the centrifugal electrospinning jet. The turbulent energy-dissipation rate model was established by using CFD(Computational Fluid Dynamics). By analyzing the electrofluid dynamic effects of the charged centrifugal jet, the transient evolution process of the jet under the influence of electric field force and centrifugal force and the driving mechanism of the field force on the jet were initially clarified. At the same time, based on the basic kinematics conditions of the jet, the jet flight partial differential equations are constructed, and the equations are solved by using matching progressively. The description of the jet two-dimensional flying deposition model and its experimental verification have been completed. The results are consistent with the results obtained from the mathematical model and can be used in the basic prediction of jet trajectories.

## Figures and Tables

**Figure 1 micromachines-13-01378-f001:**
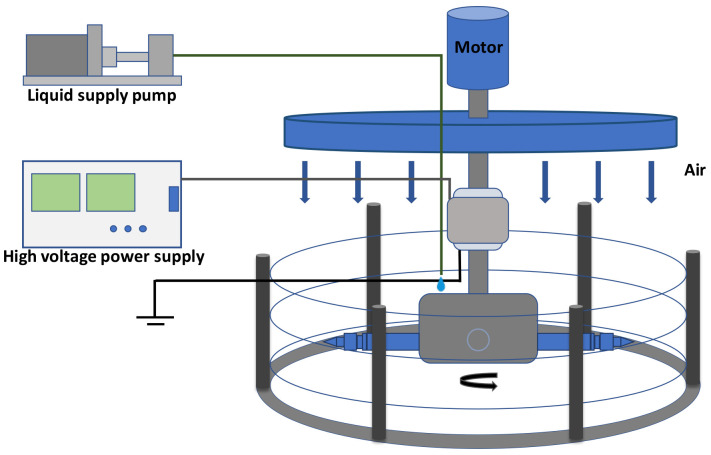
Schematic diagram of the centrifugal electrostatic jet spinning scheme.

**Figure 2 micromachines-13-01378-f002:**
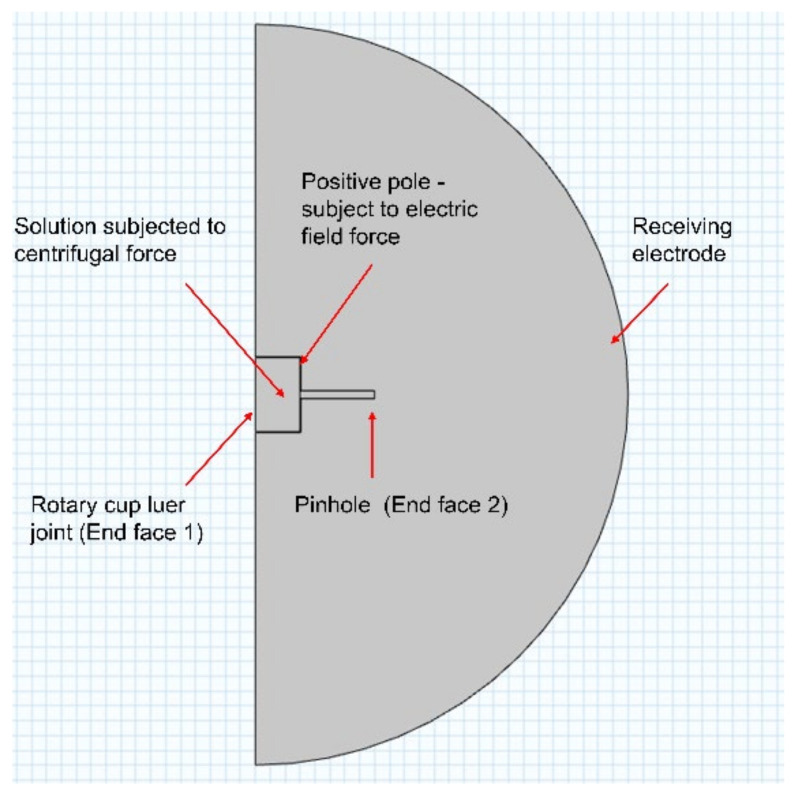
Schematic diagram of setting the boundary conditions for the simulation of the motion state of the polymer jet under the action of the centrifugal electrostatic multiphysics field.

**Figure 3 micromachines-13-01378-f003:**
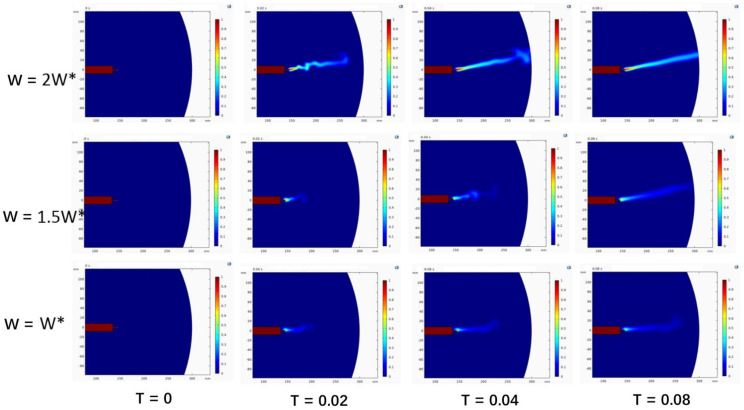
Turbulent transient integral cloud diagram of a jet at different speeds. (Note: In the figure, W* represents a dimensionless standard quantity, representing a certain rotational speed.)

**Figure 4 micromachines-13-01378-f004:**
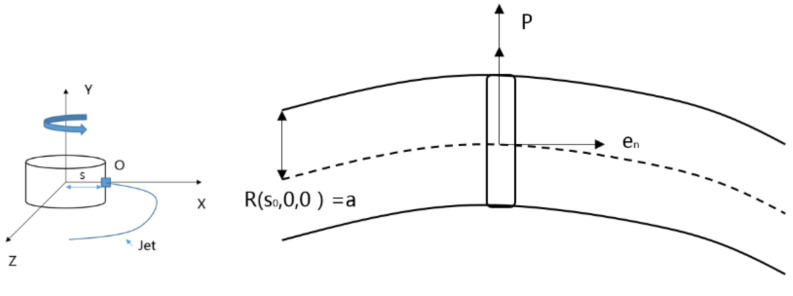
Schematic diagram of jet flow and jet cross-section in a Cartesian coordinate system.

**Figure 5 micromachines-13-01378-f005:**
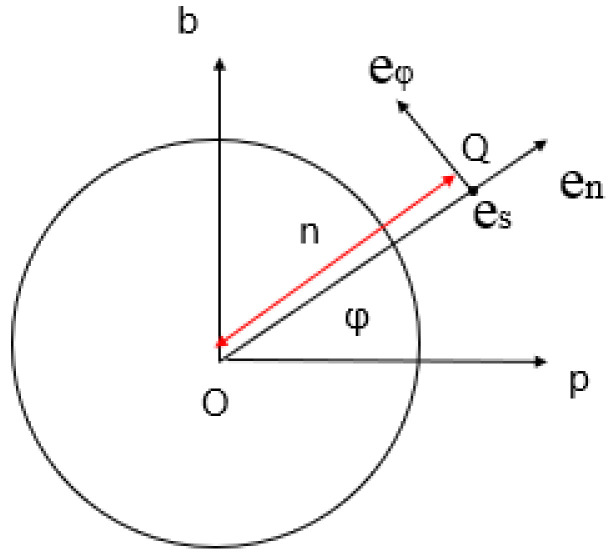
Polar coordinate diagram of jet cross-section.

**Figure 6 micromachines-13-01378-f006:**
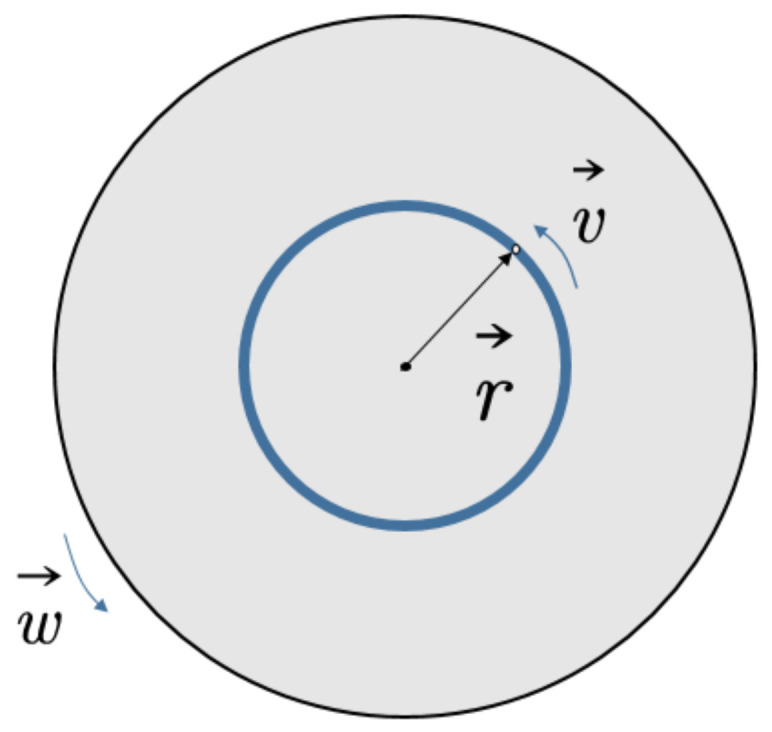
Schematic diagram of jet velocity under centrifugal action.

**Figure 7 micromachines-13-01378-f007:**
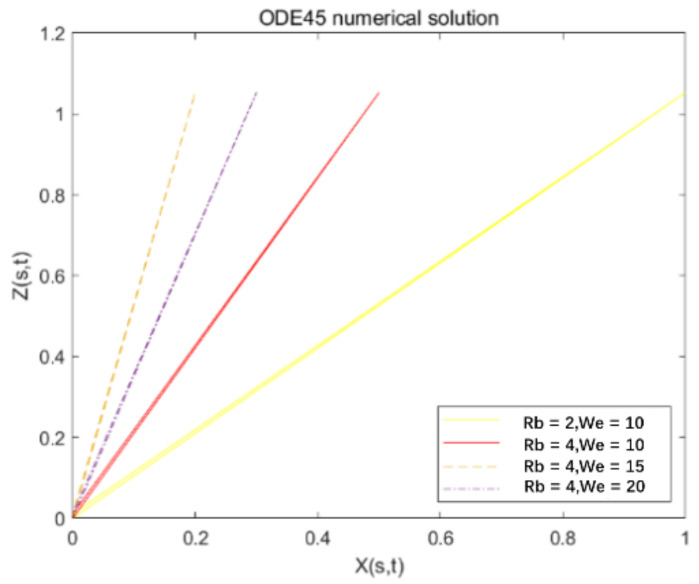
Jet trajectory diagram when the fourth-order Runge-Kutta method of Matlab(version R2016b, accessed on 7 September 2016) solves the Reynolds number and the Weber number are Re = 2, We = 10; Re = 4, We = 10, Re = 4, We = 15; Rb = 4, We = 20.

**Figure 8 micromachines-13-01378-f008:**
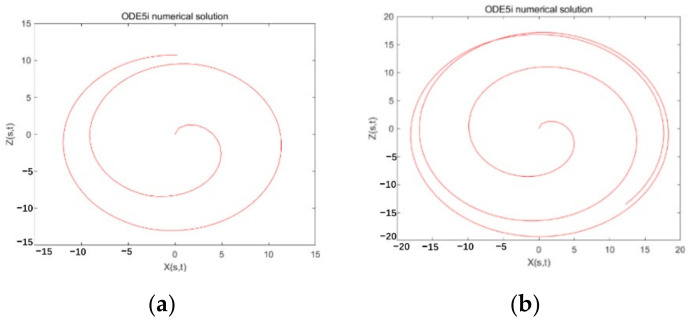
Using the implicit Runge-Kutta method ODE15I to solve the jet model. When the iteration interval is [0:300], Rb = 2, We = 10, the effect of different accuracy settings on the solution iteration. (**a**) ‘RelTol’ = 1 × 10^−3^, ‘AbsTol’ = 1 × 10^−6^; (**b**) ‘RelTol’ = 1 × 10^−4^, ‘AbsTol’ = 1 × 10^−6^; (**c**) ‘RelTol’ = 1 × 10^−5^, ‘AbsTol’ = 1 × 10^−6^; (**d**) ‘RelTol’ = 1 × 10^−6^, ‘AbsTol’ = 1 × 10^−6^.

**Figure 9 micromachines-13-01378-f009:**
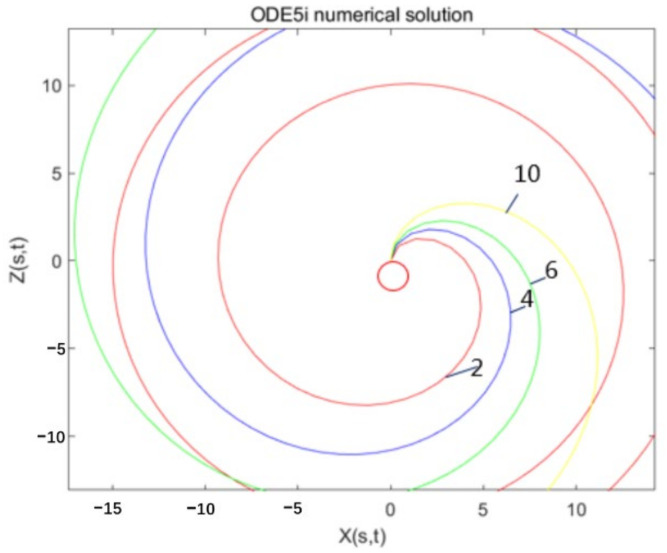
Using the implicit Runge-Kutta method ODE15i to solve the jet model. The jet trajectory diagram when We = 10, Rb = 2, 4, 6, 10 when the iteration interval is [0:300].

**Figure 10 micromachines-13-01378-f010:**
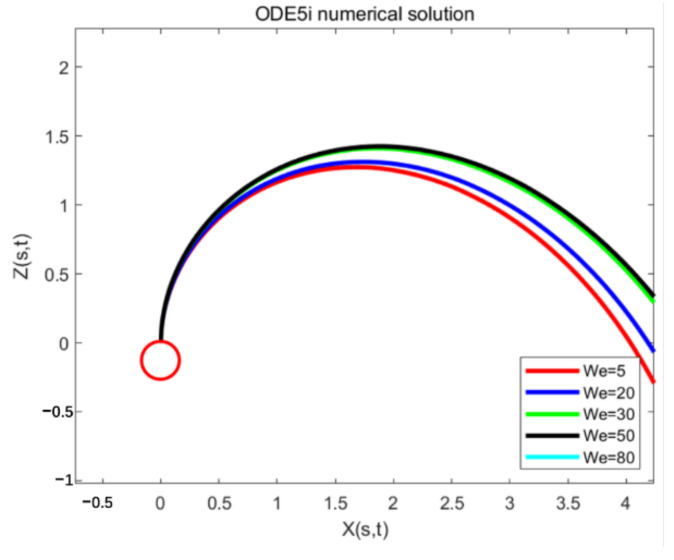
Jet trajectory diagram by using the implicit Runge-Kutta method ODE15i to solve the jet model, when the iteration interval is [0:300], the Rossby value Rb = 2, and different Weber numbers We = 5, 10, 20, 30, 50, 80.

**Figure 11 micromachines-13-01378-f011:**
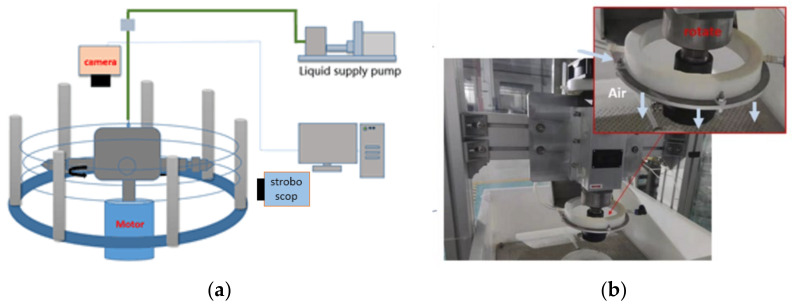
(**a**) Schematic diagram of centrifugal jet stroboscopic observation platform; (**b**) The image of centrifugal air-assisted spinning.

**Figure 12 micromachines-13-01378-f012:**
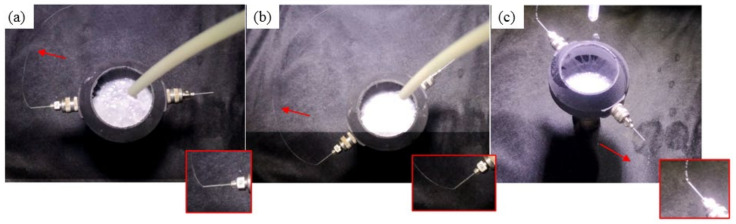
The state change of the centrifugal jet of PEO solution with different mass fractions: (**a**) 7.5%; (**b**) 6.5; (**c**) 4.5.

**Figure 13 micromachines-13-01378-f013:**
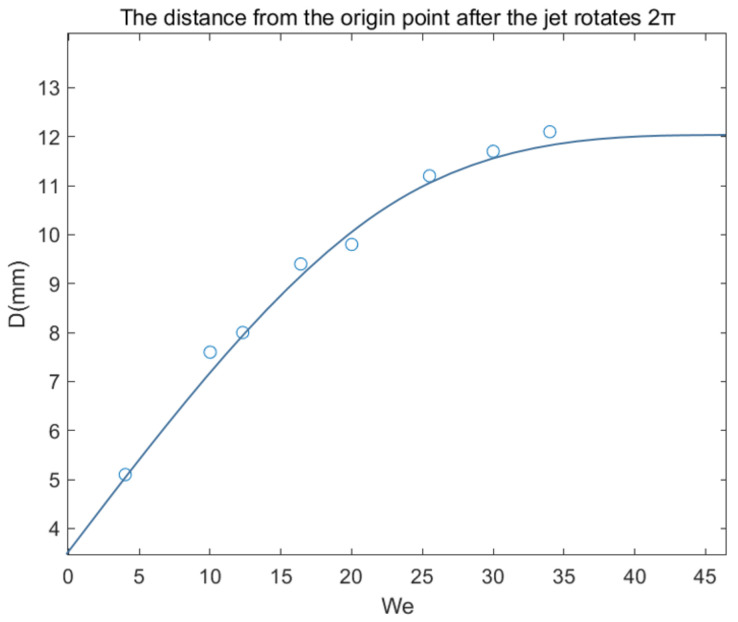
The distance from the center of the circle when the jet rotates φ = 2π.

**Table 1 micromachines-13-01378-t001:** Specifications of Centrifugal Stroboscopic Observation Platform Components.

Equipment	Manufacturers	Model Number	Specifications
Precision Gear Pump	MonyPumps (From Guangzhou, China)	M-60U2	0.001 mL/min~400 mL/min
LED Stroboscope	SuWei (From Guangzhou, China)	DW-P503-1ACDF	60–499,999 Strobe/min
			Accuracy: 0.001%
			Illuminance: 1500LUX
Microscopic camera	FASTEC IMAGING (From San Diego, CA, USA)	Hispec5	Pixel 1280 × 1024
PEO	Dow	N3000	4,000,000 g/mol

**Table 2 micromachines-13-01378-t002:** Tests of physical properties with different mass fractions.

Material	Mass Fraction	Electrical Conductivity (μs/cm)	Viscosity (mPa·s)	Surface Tension (mN/m)
PEO ^a^	4.50%	76.8	2786	49.3
PEO ^b^	5.50%	83.2	4655	52.6
PEO ^c^	6.50%	85.4	10,000	68.9
PEO ^d^	7.50%	88.6	20,200	80.3

In the table, a,b,c,d is used to identify different PEO mass fraction.

## Data Availability

Not applicable.

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
