# Peer review of "Research on the Mechanism of Multi-Domain Coupling Centrifugal Electrostatic Blowing Flying Deposition"

_micromachines, 2022, doi:10.3390/mi13091378_

Round 1

Reviewer 1 Report

In this article, the author established the turbulent energy-dissipation rate model by using CFD. They studied the electrofluid dynamic effects of the charged centrifugal jet, the transient evolution process of the jet under the influence of electric field force & centrifugal force and the driving mechanism of the field force on the jet. By using the basic kinematics conditions of the jet, they were able to construct the jet flight partial differential equations are constructed, and solved. The motivation of the paper is well grounded in the introduction. The used model is adequately described and the results are well presented.

Author Response

Thanks to the thoughtful comments on our research, we also address all reviewers' comments in the final draft as well, appreciate again to the comments. 

Reviewer 2 Report

In this manuscript, the authors proposed a centrifugal electrostatic blowing process to solve the difficult continuous and stable deposition problem in the traditional centrifugal electrostatic spinning process. Although the settings of flow channel, field force application and other parameters have been studied by many researchers, the coupling of centrifugal effect and electric field is still challenging. In this work, the authors model and analyze the flight trajectory, determine the constraint conditions and then establish a stable jet flight deposition model. CFD method is applied to simulate and analyze the electrohydrodynamic effect and explore the mechanism. Matlab is further employed to optimize the experimental conditions. The idea is quite novel and the target is clear. As a result, the actual trajectory result demonstrates consistency with the mathematical model, which can be further used for analysis and prediction. Therefore, I would like to recommend the acceptance of this manuscript after minor revision.

1. The authors should format the citations properly according to the journal requirements. 

2. Some figures can be merged to have a better arrangement. 

3. Currently, only PEO materials have been tested. Some other materials might also be included to have a better demonstration.

Author Response

  1. The authors should format the citations properly according to the journal requirements.

Citations have been properly formatted according to journal requirements.

  1. Some figures can be merged to have a better arrangement.

Some graphics have been adjusted for better alignment.

  1. Currently, only PEO materials have been tested. Some other materials might also be included to have a better demonstration.

Thanks to the judges for their suggestions. We have done experiments on centrifugal electrospinning with other materials, such as PVA materials(Zeng et al. 2021), such as DMF polymer solutions added with PVDF and silica powder(Zeng et al. 2021). The review's suggestion is very pertinent. We will continue to try other materials in the future, and there will be further scientific research reports.

References

Zeng, J., H. Wang, R. Chen, P. Wu, X. Chen, X. Chen, L. Qin, X. Lan, R. Zhang, Z. Lin, and G. Xu. (2021), “Preparation of low resistance fluffy ultrafine filter media by centrifugal electrospinning”, AIP Advances 11(9), http://dx.doi.org/10.1063/5.0051922.

Zeng, J., H. Wang, R. Chen, P. Wu, X. Chen, X. Chen, L. Qin, X. Lan, J. Guo, J. Liang, and G. Xu. (2021), ”Preparation of long-lasting electret fiber felt by centrifugal air-assisted spinning process and electret post-treatment”, AIP Advances 11(7), http://dx.doi.org/10.1063/5.0057561.